# Learning Discourse-level Diversity for Neural Dialog Models using Conditional Variational Autoencoders

## Abstract

While recent neural encoder-decoder models have shown great promise in modeling open-domain conversations, they often generate dull and generic responses. Unlike past work that has focused on diversifying the output of the decoder at word-level to alleviate this problem, we present a novel framework based on conditional variational autoencoders that captures the discourse-level diversity in the encoder. Our model uses latent variables to learn a distribution over potential conversational intents and generates diverse responses using only greedy decoders. We have further developed a novel variant that is integrated with linguistic prior knowledge for better performance. Finally, the training procedure is improved by introducing a *bag-of-word* loss. Our proposed models have been validated to generate significantly more diverse responses than baseline approaches and exhibit competence in discourse-level decision-making.

## 1 Introduction

The dialog manager is one of the key components of dialog systems, which is responsible for modeling the decision-making process. Specifically, it typically takes a new utterance and the dialog context as input, and generates discourse-level decisions (Bohus and Rudnicky, 2003; Williams and Young, 2007). Advanced dialog managers usually have a list of potential actions that enable them to have diverse behavior during a conversation, e.g. different strategies to recover from non-understanding (Yu et al., 2016). However, the conventional approach of designing a dialog manager (Williams and Young, 2007) does not

scale well to open-domain conversation models because of the vast quantity of possible decisions. Thus, there has been a growing interest in applying encoder-decoder models (Sutskever et al., 2014) for modeling open-domain conversation (Vinyals and Le, 2015; Serban et al., 2016a). The basic approach treats a conversation as a transduction task, in which the dialog history is the source sequence and the next response is the target sequence. The model is then trained end-to-end on large conversation corpora using the maximum-likelihood estimation (MLE) objective without the need for manual crafting.

However recent research has found that encoder-decoder models tend to generate generic and dull responses (e.g., *I don't know*), rather than meaningful and specific answers (Li et al., 2015; Serban et al., 2016b). There have been many attempts to explain and solve this limitation, and they can be broadly divided into two categories (see Section 2 for details): (1) the first category argues that the dialog history is only one of the factors that decide the next response. Other features should be extracted and provided to the models as conditionals in order to generate more specific responses (Xing et al., 2016; Li et al., 2016a); (2) the second category aims to improve the encoder-decoder model itself, including decoding with beam search and its variations (Wiseman and Rush, 2016), encouraging responses that have long-term payoff (Li et al., 2016b), etc.

Building upon the past work in dialog managers and encoder-decoder models, the key idea of this paper is to model dialogs as a *one-to-many* problem at the discourse level. Previous studies indicate that there are many factors in open-domain dialogs that decide the next response, and it is non-trivial to extract all of them. Intuitively, given a similar dialog history (and other observed inputs), there may exist many valid responses (at the

discourse-level), each corresponding to a certain configuration of the latent variables that are not presented in the input. To uncover the potential responses, we strive to model a probabilistic distribution over the distributed utterance embeddings of the potential responses using a latent variable (Figure 1). This allows us to generate diverse responses by drawing samples from the learned distribution and reconstruct their words via a decoder neural network.

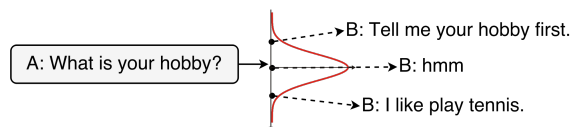

Figure 1: Given A's question, there exists many valid responses from B for different assumptions of the latent variables, e.g., B's hobby.

Specifically, our contributions are three-fold: 1. We present a novel neural dialog model adapted from conditional variational autoencoders (CVAE) (Yan et al., 2015; Sohn et al., 2015), which introduces a latent variable that can capture discourse-level variations as described above 2. We propose Knowledge-Guided CVAE (kgCVAE), which enables easy integration of expert knowledge and results in performance improvement and model interpretability. 3. We develop a training method in addressing the difficulty of optimizing CVAE for natural language generation (Bowman et al., 2015). We evaluate our models on human-human conversation data and yield promising results in: (a) generating appropriate and discourse-level diverse responses, and (b) showing that the proposed training method is more effective than the previous techniques.

## 2   Related Work

Our work is related to both recent advancement in encoder-decoder dialog models and generative models based on CVAE.

### 2.1   Encoder-decoder Dialog Models
Since the emergence of the neural dialog model, the problem of output diversity has received much attention in the research community. Ideal output responses should be both coherent and diverse. However, most models end up with generic and dull responses. To tackle this problem, one line of research has focused on augmenting the input of encoder-decoder models with richer context information, in order to generate more spe-

cific responses. Li et al., (2016a) captured speakers' characteristics by encoding background information and speaking style into the distributed embeddings, which are used to re-rank the generated response from an encoder-decoder model. Xing et al., (2016) maintain topic encoding based on Latent Dirichlet Allocation (LDA) (Blei et al., 2003) of the conversation to encourage the model to output more topic coherent responses.

On the other hand, many attempts have also been made to improve the architecture of encoder-decoder models. Li et al,. (2015) proposed to optimize the standard encoder-decoder by maximizing the mutual information between input and output, which in turn reduces generic responses. This approach penalized unconditionally high frequency responses, and favored responses that have high conditional probability given the input. Wiseman and Rush (2016) focused on improving the decoder network by alleviating the biases between training and testing. They introduced a search-based loss that directly optimizes the networks for beam search decoding. The resulting model achieves better performance on word ordering, parsing and machine translation. Besides improving beam search, Li et al., (2016b) pointed out that the MLE objective of an encoder-decoder model is unable to approximate the real-world goal of the conversation. Thus, they initialized a encoder-decoder model with MLE objective and leveraged reinforcement learning to fine tune the model by optimizing three heuristic rewards functions: informativity, coherence, and ease of answering.

### 2.2   Conditional Variational Autoencoder
The variational autoencoder (VAE) (Kingma and Welling, 2013; Rezende et al., 2014) is one of the most popular frameworks for image generation. The basic idea of VAE is to encode the input $x$ into a probability distribution $z$ instead of a point encoding in the autoencoder. Then VAE applies a decoder network to reconstruct the original input using samples from $z$. To generate images, VAE first obtains a sample of $z$ from the prior distribution, e.g. $\mathcal{N}(0, \mathbf{I})$, and then produces an image via the decoder network. A more advanced model, the conditional VAE (CVAE), is a recent modification of VAE to generate diverse images conditioned on certain attributes, e.g. generating different human faces given skin color (Yan et al., 2015; Sohn et al., 2015). Inspired by CVAE, we view the dialog contexts as the conditional attributes and adapt CVAE

to generate diverse responses instead of images.

Although VAE/CVAE has achieved impressive results in image generation, adapting this to natural language generators is non-trivial. Bowman et al., (2015) have used VAE with Long-Short Term Memory (LSTM)-based recognition and decoder networks to generate sentences from a latent Gaussian variable. They showed that their model is able to generate diverse sentences with even a greedy LSTM decoder. They also reported the difficulty of training because the LSTM decoder tends to ignore the latent variable. We refer to this issue as the *vanishing latent variable problem*. Serban et al., (2016b) have applied a latent variable hierarchical encoder-decoder dialog model to introduce utterance-level variations and facilitate longer responses. To improve upon the past models, we firstly introduce a novel mechanism to leverage linguistic knowledge in training end-to-end neural dialog models, and we also propose a novel training technique that mitigates the vanishing latent variable problem.

## 3 Proposed Models

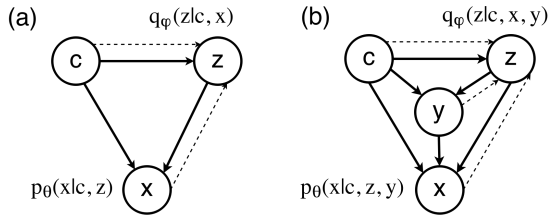

Figure 2: Graphical models of CVAE (a) and kgCVAE (b)

### 3.1 Conditional Variational Autoencoder (CVAE) for Dialog Generation

Each dyadic conversation can be represented via three random variables: the dialog context $c$ (context window size $k-1$), the response utterance $x$ (the $k^{th}$ utterance) and a latent variable $z$, which is used to capture the latent distribution over the valid responses. Further, $c$ is composed of the dialog history: the preceding k-1 utterances; conversational floor (1 if the utterance is from the same speaker of $x$, otherwise 0) and meta features $m$ (e.g. the topic). We then define the conditional distribution $p(x, z|c) = p(x|z, c)p(z|c)$ and our goal is to use deep neural networks (parametrized by $\theta$) to approximate $p(z|c)$ and $p(x|z, c)$. We refer to $p_\theta(z|c)$ as the *prior network* and $p_\theta(x, |z, c)$ as the

*response decoder*. Then the generative process of $x$ is (Figure 2 (a)):

1. Sample a latent variable $z$ from the prior network $p_\theta(z|c)$.

2. Generate $x$ through the response decoder $p_\theta(x|z, c)$.

CVAE is trained to maximize the conditional log likelihood of $x$ given $c$, which involves an intractable marginalization over the latent variable $z$. As proposed in (Sohn et al., 2015; Yan et al., 2015), CVAE can be efficiently trained with the *Stochastic Gradient Variational Bayes* (SGVB) framework (Kingma and Welling, 2013) by maximizing the variational lower bound of the conditional log likelihood. We assume the $z$ follows multivariate Gaussian distribution with a diagonal covariance matrix and introduce a *recognition network* $q_\phi(z|x, c)$ to approximate the true posterior distribution $p(z|x, c)$. Sohn and et al,. (2015) have shown that the variational lower bound can be written as:

$$\mathcal{L}(\theta, \phi; x, c) = -KL(q_\phi(z|x, c)\|p_\theta(z|c))$$
$$+ \mathbf{E}_{q_\phi(z|c,x)}[\log p_\theta(x|z, c)] \quad (1)$$
$$\leq \log p(x|c)$$

Figure 3 demonstrates an overview of our model. The utterance encoder is a bidirectional recurrent neural network (BRNN) (Schuster and Paliwal, 1997) with a gated recurrent unit (GRU) (Chung et al., 2014) to encode each utterance into fixed-size vectors by concatenating the last hidden states of the forward and backward RNN $u_i = [\vec{h_i}, \overleftarrow{h_i}]$. $x$ is simply $u_k$. The context encoder is a 1-layer GRU network that encodes the preceding k-1 utterances by taking $u_{1:k-1}$ and the corresponding conversation floor as inputs. The last hidden state $h^c$ of the context encoder is concatenated with meta features and $c = [h^c, m]$. Since we assume $z$ follows isotropic Gaussian distribution, the recognition network $q_\phi(z|x, c) \sim \mathcal{N}(\mu, \sigma^2\mathbf{I})$ and the prior network $p_\theta(z|c) \sim \mathcal{N}(\mu', \sigma'^2\mathbf{I})$, and then we have:

$$\begin{bmatrix} \mu \\ \log(\sigma^2) \end{bmatrix} = W_r \begin{bmatrix} x \\ c \end{bmatrix} + b_r \quad (2)$$

$$\begin{bmatrix} \mu' \\ \log(\sigma'^2) \end{bmatrix} = \text{MLP}_p(c) \quad (3)$$

We then use the reparametrization trick (Kingma and Welling, 2013) to obtain samples of $z$ either

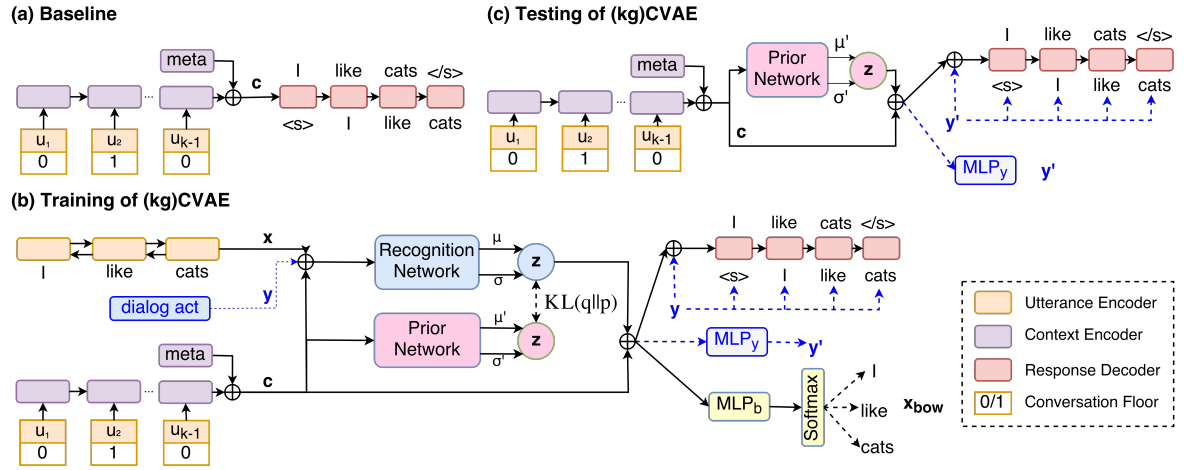

Figure 3: The neural network architectures for the baseline and the proposed CVAE/kgCVAE models. $\oplus$ denotes the concatenation of the input vectors. The dashed blue connections only appear in kgCVAE.

from $\mathcal{N}(z; \mu, \sigma^2\mathbf{I})$ predicted by the recognition network (training) or $\mathcal{N}(z; \mu', \sigma'^2\mathbf{I})$ predicted by the prior network (testing). Finally, the response decoder is a 1-layer GRU network with initial state $s_0 = W_i[z, c] + b_i$. The response decoder then predicts the words in $x$ sequentially.

### 3.2 Knowledge-Guided CVAE (kgCVAE)

In practice, training CVAE is a challenging optimization problem and often requires large amount of data. On the other hand, past research in spoken dialog systems and discourse analysis has suggested that many linguistic cues capture crucial features in representing natural conversation. For example, dialog acts (Poesio and Traum, 1998) have been widely used in the dialog managers (Litman and Allen, 1987; Raux et al., 2005; Zhao and Eskenazi, 2016) to represent the propositional function of the system. Therefore, we conjecture that it will be beneficial for the model to learn meaningful latent $z$ if it is provided with explicitly extracted discourse features during the training.

In order to incorporate the linguistic features into the basic CVAE model, we first denote the set of linguistic features as $y$. Then we assume that the generation of $x$ depends on $c$, $z$ and $y$. $y$ relies on $z$ and $c$ as shown in Figure 2. Specifically, during training the initial state of the response decoder is $s_0 = W_i[z, c, y] + b_i$ and the input at every step is $[e_t, y]$ where $e_t$ is the word embedding of $t^{th}$ word in $x$. In addition, there is an MLP to predict $y' = \text{MLP}_y(z, c)$ based on $z$ and $c$. In the testing stage, the predicted $y'$ is used by the response decoder instead of the oracle decoders. We denote the modified model as knowledge-guided

CVAE (kgCVAE) and developers can add desired discourse features that they wish the latent variable $z$ to capture. KgCVAE model is trained by maximizing:

$$\mathcal{L}(\theta, \phi; x, c, y) = -KL(q_\phi(z|x, c, y)\|P_\theta(z|c))$$
$$+ \mathbf{E}_{q_\phi(z|c,x,y)}[\log p(x|z, c, y)]$$
$$+ \mathbf{E}_{q_\phi(z|c,x,y)}[\log p(y|z, c)] \quad (4)$$

Since now the reconstruction of $y$ is a part of the loss function, kgCVAE can more efficiently encode $y$-related information into $z$ than discovering it only based on the surface-level $x$ and $c$. Another advantage of kgCVAE is that it can output a high-level label (e.g. dialog act) along with the word-level responses, which allows easier interpretation of the model's outputs.

### 3.3 Optimization Challenges

A straightforward VAE with RNN decoder fails to encode meaningful information in $z$ due to the *vanishing latent variable problem* (Bowman et al., 2015). Bowman et al., (2015) proposed two solutions: (1) *KL annealing*: gradually increasing the weight of the KL term from 0 to 1 during training; (2) *word drop decoding*: setting a certain percentage of the target words to 0. We found that CVAE suffers from the same issue when the decoder is an RNN. Also we did not consider word drop decoding because Bowman et al,. (2015) have shown that it may hurt the performance when the drop rate is too high.

As a result, we propose a simple yet novel technique to tackle the vanishing latent variable problem: *bag-of-word loss*. The idea is to introduce

an auxiliary loss that requires the decoder network to predict the bag-of-words in the response $x$ as shown in Figure 3(b). We decompose $x$ into two variables: $x_o$ with word order and $x_{bow}$ without order, and assume that $x_o$ and $x_{bow}$ are conditionally independent given $z$ and $c$: $p(x, z|c) = p(x_o|z, c)p(x_{bow}|z, c)p(z|c)$. Due to the conditional independence assumption, the latent variable is forced to capture global information about the target response. Let $f = \text{MLP}_b(z, x) \in \mathcal{R}^V$ where $V$ is vocabulary size, and we have:

$$\log p(x_{bow}|z, c) = \log \prod_{t=1}^{|x|} \frac{e^{f_{x_t}}}{\sum_j^V e^{f_j}} \qquad (5)$$

where $|x|$ is the length of $x$ and $x_t$ is the word index of $t_{th}$ word in $x$. The modified variational lower bound for CVAE with bag-of-word loss is (see Appendix A for kgCVAE):

$$\mathcal{L}'(\theta, \phi; x, c) = \mathcal{L}(\theta, \phi; x, c)$$
$$+ \mathbf{E}_{q_\phi(z|c,x,y)}[\log p(x_{bow}|z, c)] \quad (6)$$

We will show that the bag-of-word loss in Equation 6 is very effective against the vanishing latent variable and it is also complementary to the KL annealing technique.

## 4 Experiment Setup

### 4.1 Dataset

We chose the Switchboard (SW) 1 Release 2 Corpus (Godfrey and Holliman, 1997) to evaluate the proposed models. SW has 2400 two-sided telephone conversations with manually transcribed speech and alignment. In the beginning of the call, a computer operator gave the callers recorded prompts that define the desired topic of discussion. There are 70 available topics. We randomly split the data into 2316/60/62 dialogs for train/validate/test. The pre-processing includes (1) tokenize using the NLTK tokenizer (Bird et al., 2009); (2) remove non-verbal symbols and repeated words due to false starts; (3) keep the top 10K frequent word types as the vocabulary. The final data have $207, 833/5, 225/5, 481$ $(c, x)$ pairs for train/validate/test. Furthermore, a subset of SW was manually labeled with dialog acts (Stolcke et al., 2000). We extracted dialog act labels based on the dialog act recognizer proposed in (Ribeiro et al., 2015). The features include the uni-gram and bi-gram of the utterance, and the contextual features of the last 3 utterances. We trained a Support Vector Machine

(SVM) (Suykens and Vandewalle, 1999) with linear kernel on the subset of SW with human annotations. There are 42 types of dialog acts and the SVM achieved 77.3% accuracy on held-out data. Then the rest of SW data are labelled with dialog acts using the trained SVM dialog act recognizer.

### 4.2 Training

We trained with the following hyperparameters (according to the loss on the validate dataset): word embedding has size 200 and is shared across everywhere. We initialize the word embedding from Glove embedding pre-trained on Twitter (Pennington et al., 2014). The utterance encoder has a hidden size of 300 for each direction. The context encoder has a hidden size of 600 and the response decoder has a hidden size of 400. The prior network and the MLP for predicting $y$ both have 1 hidden layer of size 400 and $tanh$ non-linearity. The latent variable $z$ has a size of 200. The context window $k$ is 10. All the initial weights are sampled from a uniform distribution [-0.08, 0.08]. The mini-batch size is 30. The models are trained end-to-end using the Adam optimizer (Kingma and Ba, 2014) with a learning rate of 0.001 and gradient clipping at 5. We selected the best models based on the variational lower bound on the validate data. Finally, we use the BOW loss along with *KL annealing* of 10,000 batches to achieve the best performance. Section 5.4 gives a detailed argument for the importance of the BOW loss.

## 5 Results

### 5.1 Experiments Setup

We compared three neural dialog models: a strong baseline model, CVAE, and kgCVAE. The **baseline model** is an encoder-decoder neural dialog model without latent variables similar to (Serban et al., 2016a). The baseline model's encoder uses the same context encoder to encode the dialog history and the meta features as shown in Figure 3. The encoded context $c$ is directly fed into the decoder networks as the initial state. The hyperparameters of the baseline are the same as the ones reported in Section 4.2 and the baseline is trained to minimize the standard cross entropy loss of the decoder RNN model without any auxiliary loss.

Also, to compare the diversity introduced by the stochasticity in the proposed latent variable versus the softmax of RNN at each decoding step, we generate $N$ responses from the baseline by sam-

pling from the softmax. For CVAE/kgCVAE, we sample $N$ times from the latent $z$ and only use greedy decoders so that the randomness comes entirely from the latent variable $z$.

## 5.2 Quantitative Analysis

Automatically evaluating an open-domain generative dialog model is an open research challenge (Liu et al., 2016). Following our *one-to-many* hypothesis, we propose the following metrics. We assume that for a given dialog context $c$, there exist $M_c$ reference responses $r_j$, $j \in [1, M_c]$. Meanwhile a model can generate $N$ hypothesis responses $h_i$, $i \in [1, N]$. The generalized response-level precision/recall for a given dialog context is:

$$\text{precision}(c) = \frac{\sum_{i=1}^{N} max_{j \in [1, M_c]} d(r_j, h_i)}{N}$$

$$\text{recall}(c) = \frac{\sum_{j=1}^{M_c} max_{i \in [1, N]} d(r_j, h_i))}{M_c}$$

where $d(r_j, h_i)$ is a distance function which lies between 0 to 1 and measures the similarities between $r_j$ and $h_i$. The final score is averaged over the entire test dataset and we report the performance with 3 types of distance functions in order to evaluate the systems from various linguistic points of view:

1. Smoothed Sentence-level BLEU (Chen and Cherry, 2014): BLEU is a popular metric that measures the geometric mean of modified n-gram precision with a length penalty (Papineni et al., 2002; Li et al., 2015). We use BLEU-1 to 4 as our lexical similarity metric.

2. Cosine Distance of Bag-of-word Embedding: a simple method to obtain sentence embeddings is to take the average or extrema of all the word embeddings in the sentences (Forgues et al., 2014; Adi et al., 2016). The $d(r_j, h_i)$ is the cosine distance of the two embedding vectors. We used Glove embedding described in Section 4 and denote the average method as A-bow and extrema method as E-bow.

3. Dialog Act Match: to measure the similarity at the discourse level, the same dialog-act tagger from 4.1 is applied to label all the generated responses of each model. We set $d(r_j, h_i) = 1$ if $r_j$ and $h_i$ have the same dialog acts, otherwise $d(r_j, h_i) = 0$.

One challenge of using the above metrics is that there is only one, rather than multiple reference responses/contexts. This impacts reliability of our measures. Inspired by (Sordoni et al., 2015), we utilized information retrieval techniques (see Appendix A) to gather 10 extra candidate reference responses/context from other conversations with the same topics. Then the 10 candidate references are filtered by two experts, which serve as the ground truth to train the reference response classifier. The result is 6.69 references in average per context. The average number of distinct reference dialog acts is 4.2. Table 1 shows the results.

| Metrics | Baseline | CVAE | kgCVAE |
|---|---|---|---|
| perplexity (KL) | 35.4 | 20.2 | 16.02 |
|  | (n/a) | (11.36) | (13.08) |
| BLEU-1 prec | 0.405 | 0.372 | **0.412** |
| BLUE-1 recall | 0.336 | 0.381 | **0.411** |
| BLEU-2 prec | 0.300 | 0.295 | **0.310** |
| BLEU-2 recall | 0.281 | **0.322** | 0.318 |
| BLEU-3 prec | 0.272 | 0.265 | **0.310** |
| BLEU-3 recall | 0.254 | 0.292 | **0.318** |
| BLEU-4 prec | 0.226 | 0.223 | **0.262** |
| BLEU-4 recall | 0.215 | 0.248 | **0.272** |
| A-bow prec | 0.387 | **0.389** | 0.373 |
| A-bow recall | 0.337 | **0.361** | 0.336 |
| E-bow prec | 0.701 | 0.705 | **0.711** |
| E-bow recall | 0.684 | 0.709 | **0.712** |
| DA prec | **0.736** | 0.704 | 0.721 |
| DA recall | 0.514 | **0.604** | 0.598 |

Table 1: Performance of each model on automatic measures. The highest score in each row is in bold.

Both CVAE and kgCVAE outperform the baseline in terms of recall in all the metrics. This confirms our hypothesis that generating responses with discourse-level diversity can lead to a more comprehensive coverage of the potential responses than promoting only word-level diversity. As for precision, we observed that the baseline has higher or similar scores than CVAE in all metrics, which is expected since the baseline tends to generate the mostly likely and safe responses repeatedly in the $N$ hypotheses. However, kgCVAE is able to achieve the highest precision and recall in the 4 metrics at the same time (BLEU1/3/4, E-BOW). One reason for kgCVAE's good performance is that the predicted dialog act label in kgCVAE can regularize the generation process of its RNN decoder by forcing it to generate more coherent

and precise words. We further analyze the precision/recall of BLEU-4 by looking at the average score versus the number of distinct reference dialog acts. A low number of distinct dialog acts represents the situation where the dialog context has a strong constraint on the range of the next response (low entropy), while a high number indicates the opposite (high-entropy). Figure 4 shows that CVAE/kgCVAE achieves significantly higher recall than the baseline in higher entropy contexts. Also it shows that CVAE suffers from lower precision, especially in low entropy contexts. Finally, kgCVAE gets higher precision than both the baseline and CVAE in the full spectrum of context entropy.

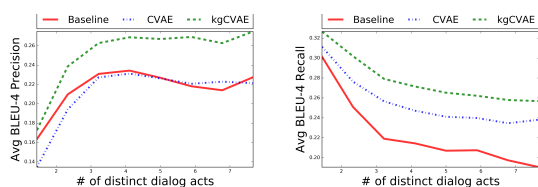

Figure 4: BLEU-4 precision/recall vs. the number of distinct reference dialog acts.

## 5.3 Qualitative Analysis

Table 2 shows the outputs generated from the baseline and kgCVAE. In example 1, caller A begins with an open-ended question. The kgCVAE model generated highly diverse answers that cover multiple plausible dialog acts. Further, we notice that the generated text exhibits similar dialog acts compared to the ones predicted separately by the model, implying the consistency of natural language generation based on $y$. On the contrary, the responses from the baseline model are limited to local n-gram variations and share a similar prefix, i.e. "I'm". Example 2 is a situation where caller A is telling B stories. The ground truth response is a back-channel and the range of valid answers is more constrained than example 1 since B is playing the role of a listener. The baseline successfully predicts "uh-huh". The kgCVAE model is also able to generate various ways of back-channeling. This implies that the latent $z$ is able to capture context-sensitive variations, i.e. in low-entropy dialog contexts modeling lexical diversity while in high-entropy ones modeling discourse-level diversity. Moreover, kgCVAE is occasionally able to generate more sophisticated grounding (sample 4) beyond a simple back-channel, which is also an acceptable response given the dialog context.

In addition, past work (Kingma and Welling, 2013) has shown that the recognition network is able to learn to cluster high-dimension data, so we conjecture that posterior $z$ outputted from the recognition network should cluster the responses into meaningful groups. Figure 5 visualizes the posterior $z$ of responses in the test dataset in 2D space using t-SNE (Maaten and Hinton, 2008). We found that the learned latent space is highly correlated with the dialog act and length of responses, which confirms our assumption.

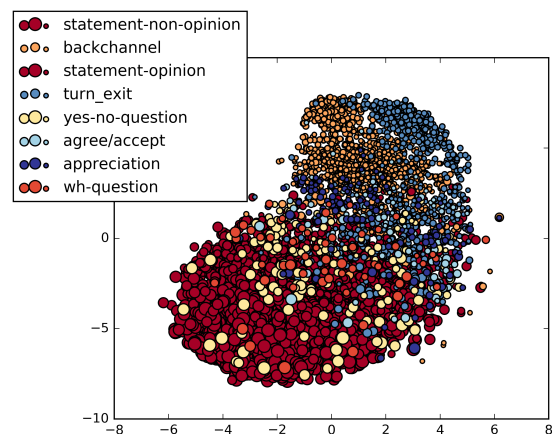

Figure 5: t-SNE visualization of the posterior $z$ for test responses with top 8 frequent dialog acts. The size of circle represents the response length.

## 5.4 Results for Bag-of-Word Loss

Finally, we evaluate the effectiveness of bag-of-word (BOW) loss for training VAE/CVAE with the RNN decoder. To compare with past work (Bowman et al., 2015), we conducted the same language modelling (LM) task on Penn Treebank using VAE. The network architecture is same except we use GRU instead of LSTM. We compared four different training setups: (1) standard VAE without any heuristics; (2) VAE with KL annealing (KLA); (3) VAE with BOW loss; (4) VAE with both BOW loss and KLA. Intuitively, a well trained model should lead to a low reconstruction loss and small but non-trivial KL cost. For all models with KLA, the KL weight increases linearly from 0 to 1 in the first 5000 batches.

Table 3 shows the reconstruction perplexity and the KL cost on the test dataset. The standard VAE fails to learn a meaningful latent variable by having a KL cost close to 0 and a reconstruction perplexity similar to a small LSTM LM (Zaremba et al., 2014). KLA helps to improve the recon-

**Example 1-Topic:** Recycling **Context:** A: are they doing a lot of recycling out in Georgia? **Target-B** (statement): well at my workplace we have palaces for aluminium cans and we have a separate trash can for recyclable paper

| Baseline+Sampling | kgCVAE+Greedy |
|---|---|
| 1. well I'm a graduate student and have two kids | 1. (non-understand) pardon |
| 2. well I was in last year and so we've had lots of recycling | 2. (statement) oh you're not going to have a curbside pick up here |
| 3. I'm not sure | 3. (statement) okay I am sure about a recycling center |
| 4. well I don't know I just moved here in new york | 4. (yes-answer) yeah so |

**Example 2-Topic:** Child Care **Context:** A: you know a private home to take their children to when they're young until they hit the preschool age and they **Target-B** (backchannel): uh-huh

| Baseline+Sampling | kgCVAE+Greedy |
|---|---|
| 1. um - hum | 1. (backchannel) uh-huh |
| 2. yeah | 2. (turn-exit) um-hum |
| 3. um - hum | 3. (backchannel) yeah |
| 4. uh-huh | 4. (statement) oh yeah I think that's part of the problem |

Table 2: Generated responses from the baselines and kgCVAE in two examples. KgCVAE also provides the predicted dialog act for each response. The context only shows the last utterance due to space limit (the actual context window size is 10).

struction loss, but it requires early stopping since the models will fall back to the standard VAE after the KL weight becomes 1. At last, the models with BOW loss achieved significantly lower perplexity and larger KL cost.

| Model | Perplexity | KL cost |
|---|---|---|
| Standard | 122.0 | 0.05 |
| KLA | 111.5 | 2.02 |
| BOW | 97.72 | 7.41 |
| BOW+KLA | 73.04 | 15.94 |

Table 3: The reconstruction perplexity and KL terms on Penn Treebank test set.

Figure 6 visualizes the evolution of the KL cost. We can see that for the standard model, the KL cost crashes to 0 at the beginning of training and never recovers. On the contrary, the model with only KLA learns to encode substantial information in latent $z$ when the KL cost weight is small. However, after the KL weight is increased to 1 (after 5000 batch), the model once again decides to ignore the latent $z$ and falls back to the naive implementation. The model with BOW loss, however, consistently converges to a non-trivial KL cost even without KLA, which confirms the importance of BOW loss for training latent variable models with the RNN decoder. Last but not least, our experiments showed that the conclusions drawn from LM using VAE also apply to training CVAE/kgCVAE, so we used BOW loss together with KLA for all previous experiments.

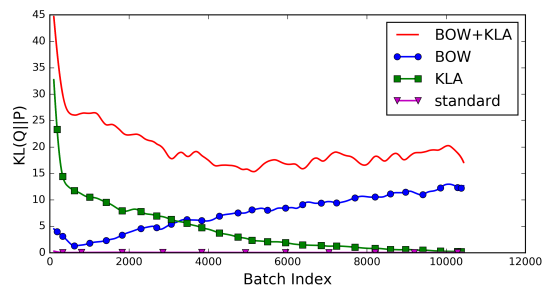

Figure 6: The value of the KL divergence during training with different setups on Penn Treebank.

# 6 Conclusion and Future Work

In conclusion, we identified the *one-to-many* nature of open-domain conversation and proposed two novel models that show superior performance in generating diverse and appropriate responses at the discourse level. While the current paper addresses diversifying responses in respect to dialogue acts, this work is part of a larger research direction that targets leveraging both past linguistic findings and the learning power of deep neural networks to learn better representation of the latent factors in dialog. In turn, the output of this novel neural dialog model will be easier to explain and control by humans. In addition to dialog acts, we plan to apply our kgCVAE model to capture other different linguistic phenomena including sentiment, named entities, etc. Last but not least, the recognition network in our model will serve as the foundation for designing a data-driven dialog manager, which automatically discovers useful high-level intents. All of the above suggest a promising research direction.

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

## A  Supplemental Material

### Variational Lower Bound for kgCVAE

We assume that even with the presence of linguistic feature $y$ regarding $x$, the prediction of $x_{bow}$ still only depends on the $z$ and $c$. Therefore, we have:

$$
\begin{aligned}
\mathcal{L}(\theta, \phi; x, c, y) = &-KL(q_\phi(z|x,c,y)\|P_\theta(z|c)) \\
&+ \mathbf{E}_{q_\phi(z|c,x,y)}[\log p(x|z,c,y)] \\
&+ \mathbf{E}_{q_\phi(z|c,x,y)}[\log p(y|z,c)] \\
&+ \mathbf{E}_{q_\phi(z|c,x,y)}[\log p(x_{bow}|z,c)]
\end{aligned}
\tag{7}
$$

### Collection of Multiple Reference Responses

We collected multiple reference responses for each dialog context in the test set by information retrieval techniques combining with traditional a machine learning method. First, we encode the dialog history using Term Frequency-Inverse Document Frequency (TFIDF) (Salton and Buckley, 1988) weighted bag-of-words into vector representation $h$. Then we denote the topic of the conversation as $t$ and denote $f$ as the conversation floor, i.e. if the speakers of the last utterance in the dialog history and response utterance are the same $f = 1$ otherwise $f = 0$. Then we computed the similarity $d(c_i, c_j)$ between two dialog contexts using:

$$
d(c_i, c_j) = \mathbb{1}(t_i = t_j)\mathbb{1}(t_i = t_j)\frac{h_i \cdot h_j}{||h_i||||h_j||} \quad (8)
$$

Unlike past work (Sordoni et al., 2015), this similarity function only cares about the distance in the context and imposes no constraints on the response, therefore is suitbale for finding diverse responses regarding to the same dialog context. Secondly, for each dialog context in the test set, we retrieved the 10 nearest neighbors from the training set and treated the responses from the training set as candidate reference responses. Thirdly, we further sampled 240 context-responses pairs from 5481 pairs in the total test set and post-processed the selected candidate responses by two human computational linguistic experts who were told to give a binary label for each candidate response about whether the response is appropriate regarding its dialog context. The filtered lists then served as the ground truth to train our reference response classifier. For the next step, we extracted bigrams, part-of-speech bigrams and word part-of-speech

pairs from both dialogue contexts and candidate reference responses with rare threshold for feature extraction being set to 20. Then L2-regularized logistic regression with 10-fold cross validation was applied as the machine learning algorithm. Cross validation accuracy on the human-labelled data was 71%. Finally, we automatically annotated the rest of test set with this trained classifier and the resulting data were used for model evaluation.

