# Peer review of "Learning Discourse-level Diversity for Neural Dialog Models using Conditional Variational Autoencoders"

_ACL 2017 — decision unknown_

[Official Review · Reviewer 1 · rating 4 · confidence 4]
soundness 5 · originality 5 · clarity 5 · impact 3 · substance 4 · appropriateness 5 · meaningful comparison 3 · presentation format Oral Presentation

Review, ACL 2017, paper 256:

This paper extends the line of work which models generation in dialogue as a
sequence to sequence generation problem, where the past N-1 utterances (the
‘dialogue context’) are encoded into a context vector (plus potential
other, hand-crafted features), which is then decoded into a response: the Nth
turn in the dialogue. As it stands, such models tend to suffer from lack of
diversity, specificity and local coherence in the kinds of response they tend
to produce when trained over large dialogue datasets containing many topics
(e.g. Cornell, Opensubtitles, Ubuntu, etc.). Rather than attempting to produce
diverse responses using the decoder, e.g. through word-by-word beam search
(which has been shown not to work very well, even lose crucial information
about grammar and valid sequences), or via a different objective function (such
as in Li et. al.’s work) the authors introduce a latent variable, z, over
which a probability distribution is induced as part of the network. At
prediction time, after encoding utterances 1 to k, a context z is sampled, and
the decoder is greedily used to generate a response from this. The evaluation
shows small improvements in BLEU scores over a vanilla seq2seq model that does
not involve learning a probability distribution over contexts and sampling from
this.

The paper is certainly impressive from a technical point of view, i.e. in the
application of deep learning methods, specifically conditioned variational auto
encoders, to the problem of response generation, and its attendant difficulties
in training such models. Their use of Information-Retrieval techniques to get
more than one reference response is also interesting. 

I have some conceptual comments on the introduction and the motivations behind
the work, some on the model architecture, and the evaluation which I write
below in turn:

Comments on the introduction and motivations…. 

The authors seem not fully aware of the long history of this field, and its
various facets, whether from a theoretical perspective, or from an applied one.

1. “[the dialogue manager] typically takes a new utterance and the dialogue
context as input, and generates discourse level decisions.” 

        This is not accurate. Traditionally at least, the job of the dialogue
manager is to select actions (dialogue acts) in a particular dialogue context.
The                    action chosen is then passed to a separate generation
module
for
realisation. Dialogue management is usually done in the context of task-based
systems which are goal driven. The dialogue manager is to choose actions which
are optimal in some sense, e.g. reach a goal (e.g. book a restaurant) in as few
steps as possible. See publications from Lemon & Pietquin, 2012, Rieser, Keizer
and colleagues, and various publications from Steve Young, Milica Gasic and
colleagues for an overview of the large literature on Reinforcement Learning
and MDP models for task-based dialogue systems.

2. The authors need to make a clear distinction between task-based,
goal-oriented dialogue, and chatbots/social bots, the latter being usually no
more than a language model, albeit a sophisticated one (though see Wen et. al.
2016). What is required from these two types of system is usually distinct.
Whereas the former is required to complete a task, the latter is, perhaps only
required to keep the user engaged. Indeed the data-driven methods that have
been used to build such systems are usually very different. 
3. The authors refer to ‘open-domain’ conversation. I would suggest that
there is no such thing as open-domain conversation - conversation is always in
the context of some activity and for doing/achieving something specific in the
world. And it is this overarching goal, the overarching activity, this
overarching genre, which determines the outward shape of dialogues and
determines what sorts of dialogue structure are coherent. Coherence itself is
activity/context-specific. Indeed a human is not capable of open-domain
dialogue: if they are faced with a conversational topic or genre that they have
never participated in, they would embarrass themselves with utterances that
would look incoherent and out of place to others already familiar with it.
(think of a random person on the street trying to follow the conversations at
some coffee break at ACL). This is the fundamental problem I see with systems
that attempt to use data from an EXTREMELY DIVERSE, open-ended set of
conversational genres (e.g. movie subtitles) in order to train one model,
mushing everything together so that what emerges at the other end is just very
good grammatical structure. Or very generic responses. 

Comments on the model architecture:

Rather than generate from a single encoded context, the authors induce a
distribution over possible contexts, sample from this, and generate greedily
with the decoder. It seems to me that this general model is counter intuitive,
and goes against evidence from the Linguistic/Psycholinguistic literature on
dialogue: this literature shows that people tend to resolve potential problems
in understanding and acceptance very locally - i.e. make sure they agree on
what the context of the conversation is - and only then move on with the rest
of the conversation, so that at any given point, there is little uncertainty
about the current context of the conversation. The massive diversity one sees
results from the diversity in what the conversation is actually trying to
achieve (see above), diversity in topics and contexts etc, so that in a given,
fixed context, there is a multitude of possible next actions, all coherent, but
leading the conversation down a different path.

It therefore seems strange to me at least to shift the burden of explaining
diversity and coherence in follow-up actions to that of the
linguistic/verbal/surface contexts in which they are uttered, though of course,
uncertainty here can also arise as a result of mismatches in vocabulary,
grammars, concepts, people’s backgrounds etc. But this probably wouldn’t
explain much of the variation in follow-up response. 

In fact, at least as far as task-based Dialogue systems are concerned, the
challenge is to capture synonymy of contexts, i.e. dialogues that are distinct
on the surface, but lead to the same or similar context, either in virtue of
interactional and syntactic equivalence relations, or synonymy relations that
might hold in a particular domain between words or sequences of words (e.g.
“what is your destination?” = “where would you like to go?” in a flight
booking domain). See e.g. Bordes & Weston, 2016; and Kalatzis, Eshghi & Lemon,
2016 - the latter use a grammar to cluster semantically similar dialogues.

Comments on the evaluation:

The authors seek to show that their model can generate more coherent, and more
diverse responses. The evaluation method, though very interesting, seems to
address coherence but not diversity, despite what they say in section 5.2:

The precision and recall metrics measure distance between ground truth
utterances and the ones the model generates, but not that between the generated
utterances themselves (unless I’m misunderstanding the evaluation method).
See e.g. Li et al. who measure diversity by counting the number distinct
n-grams in the generated responses.

Furthermore, I’m not sure that the increase in BLEU scores are meaningful:
they are very small. In the qualitative assessment of the generated responses,
one certainly sees more diversity, and more contentful utterances in the
examples provided. But I can’t see how frequent such cases in fact are.

Also, it would have made for a stronger, more meaningful paper if the authors
had compared their results with other work, (e.g. Li et. al) that use very
different methods to promote diversity (e.g. by using a different objective
function). The authors in fact do not mention this, or characterise it
properly, despite actually referring to Li et. al. 2015.

[Official Review · Reviewer 2 · rating 5 · confidence 4]
soundness 5 · originality 5 · clarity 4 · impact 3 · substance 5 · appropriateness 5 · meaningful comparison 3 · presentation format Oral Presentation

This paper presents a neural sequence-to-sequence model for encoding dialog
contexts followed by decoding system responses in open-domain conversations.
The authors introduced conditional variational autoencoder (CVAE) which is a
deep neural network-based generative model to learn the latent variables for
describing responses conditioning dialog contexts and dialog acts.
The proposed models achieved better performances than the baseline based on RNN
encoder-decoder without latent variables in both quantitative and qualitative
evaluations.

This paper is well written with clear descriptions, theoretically sound ideas,
reasonable comparisons, and also detailed analysis.
I have just a few minor comments as follows:

- Would it be possible to provide statistical significance of the results from
the proposed models compared to the baseline in quantitative evaluation? The
differences don't seem that much for some metrics.

- Considering the importance of dialog act in kgCVAE model, the DA tagging
performances should affect the quality of the final results. Would it be there
any possibility to achieve further improvement by using better DA tagger?
Recently, deep learning models have achieved better performances than SVM also
in DA tagging.

- What do you think about doing human evaluation as a part of qualitative
analysis? It could be costly, but worth a try to analyze the results in more
pragmatic perspective.

- As a future direction, it could be also interesting if kgCVAE model is
applied to more task-oriented human-machine conversations which usually have
much richer linguistic features available than open conversation.

- In Table 1, 'BLUE-1 recall' needs to be corrected to 'BLEU-1 recall'.